# Correlation between Pitch Impregnation Pressure and Pore Sizes of Graphite Block

**DOI:** 10.3390/ma15020561

**Published:** 2022-01-12

**Authors:** Changkyu Kim, Woong Kwon, Moon Hee Lee, Jong Seok Woo, Euigyung Jeong

**Affiliations:** 1Department of Textile System Engineering, Kyungpook National University, Daegu 41566, Korea; se02126000@gmail.com (C.K.); kwoong7242@naver.com (W.K.); moonhee.lee@morganplc.com (M.H.L.); 2Advanced Center of Engineering, Morgan Advanced Materials, 23, Dalseong2cha 4-ro, Guji-myeon, Dalseong-gun, Daegu 43013, Korea; JustinJongSeok.Woo@morganplc.com

**Keywords:** artificial graphite, graphite block, pitch, impregnation, pore properties

## Abstract

This study aimed to investigate the effect of impregnation pressure on the decrease in porosity of impregnated bulk graphite. The correlation between pitch impregnation behavior and the pore sizes of the bulk graphite block was studied to determine the optimal impregnation pressure. The densities and porosities of the bulk graphite before and after pitch impregnation under various pressures between 10 and 50 bar were evaluated based on the Archimedes method and a mercury porosimeter. The density increased rates increased by 1.93–2.44%, whereas the impregnation rate calculated from the rate of open porosity decreased by 15.15–24.48%. The density increase rate and impregnation rate were significantly high when the impregnation pressures were 40 and 50 bar. Compared with impregnation pressures of 10, 20, and 30 bar, the minimum impregnatable pore sizes with impregnation pressures of 40 and 50 bar were 30–39 and 24–31 nm, respectively. The mercury intrusion porosimeter analysis results demonstrated that the pressure-sensitive pore sizes of the graphite blocks were in the range of 100–4500 nm. Furthermore, the ink-bottle-type pores in this range contributed predominantly to the effect of impregnation under pressure, given that the pitch-impregnated-into-ink-bottle-type pores were difficult to elute during carbonization.

## 1. Introduction

Artificial graphite is widely used in various industries due to its high thermal and electrical conductivity, chemical resistance, high mechanical strength, low thermal expansion, excellent thermal shock resistance under rapid temperature changes, de-wetting, self-lubrication, air-tightness, and machinability [1,2,3]. There are two forms of artificial graphite: powder and bulk. Artificial graphite powders, such as graphene and graphite/graphene composites, are used primarily as anode materials for secondary ion batteries (lithium ion, sodium ion, and aluminum ion) and supercapacitors. By contrast, artificial bulk graphite is used in silicon ingots for semiconductors, nuclear reactor moderators, bearings, special machine parts, mechanical seals, cell divider plates, and steel-making electrode bars [4,5,6,7,8,9].

The general manufacturing process of bulk graphite starts with mixing petroleum- or coal-tar-pitch-derived cokes with a binder pitch or resin. The mixture is then manufactured into bulk graphite through molding, carbonization, impregnation, graphitization, and purification [10,11,12]. When the molded mixture is carbonized, volatile compounds are released from the pyrolysis or evaporation of a binder pitch or resin components, generating pores with various sizes inside the bulk graphite that degrade the graphite’s mechanical, electrical, and thermal properties [12]. Consequently, impregnation followed by heat treatment is required to minimize these pores to produce the desired properties of the bulk graphite.

The viscosity, surface tension, contact angle, and carbon yield during impregnation are crucial for minimizing the porosity of bulk graphite [13,14,15,16,17,18]. The impregnation conditions, such as pressure and temperature profile, are also critical [3,19]. Lee et al. reported that the impregnation of phenol resin increased as its viscosity decreased and that the impregnation was affected by the porosity of the graphite. As the porosity of the graphite increased, the impregnation rate increased [13,14]. Although other parameters affect the impregnation of the bulk graphite, related studies are scarce. The impregnation temperature and pressure are of practical relevance because they can influence the economics of impregnation. When higher temperatures and pressures are required, the cost of impregnation dramatically increases.

Accordingly, this study aimed to investigate the effect of impregnation pressure on the decrease in porosity of impregnated bulk graphite. The correlation between the pitch impregnation behaviors and pore sizes of bulk graphite blocks was studied to determine the optimal impregnation pressure. The apparent densities and porosities of the bulk graphite before and after pitch impregnation under various pressures between 10 and 50 bar were evaluated based on the Archimedes method and a mercury porosimeter.

## 2. Materials and Methods

### 2.1. Materials

Coal tar pitch was used as an impregnator and purchased from Ruetgers Germany GmbH (Castrop-Rauxel, Germany). Bulk graphite blocks were provided by the Morgan Korea company (Daegu, Korea). Diiodomethane (99+%) was purchased from Acros Organics (Waltham, MA, USA).

### 2.2. Characterization of the Impregnation Pitch

The elemental contents of the impregnation pitch were analyzed using an elemental analyzer (Flash 2000, Thermo Fisher Scientific, Waltham, MA, USA). Carbon yield was measured using thermogravimetric analysis (TGA, Q 500, TA instrument, New Castle, DE, USA) from 25 to 1000 °C with a heating rate of 5 °C/min under a nitrogen atmosphere. The softening point of the impregnation pitch was evaluated based on ASTM D3416 with a heating rate of 2 °C/min using a softening point analyzer (DP-70, Mettler Toledo, Columbus, OH, USA). The contents of the quinoline insoluble were measured based on ASTM D2318, and the contents of toluene insoluble were measured based on ASTM D4312. The properties of the pitch impregnator are presented in Table 1.

### 2.3. Impregnation of Bulk Graphite Block

Graphite blocks were dried at 110 °C for 8 h in a convection oven before impregnation. The dried block and pitch were placed in the reactor of the impregnating apparatus, and the reactor was vacuumed for 30 min to remove any residual water vapor inside the graphite block and pitch. The reactor was then heated to 110 °C to melt the pitch, and the designated pressure was applied for impregnation of the pitch into the graphite blocks. The pressures used in the impregnation were 10, 20, 30, 40, and 50 bar. This condition was maintained for 1 h, and the reactor was cooled and depressurized to room temperature and pressure. The impregnated block was removed from the reactor and carbonized at 1000 °C for 1 h. After carbonization of the impregnated block, the eluted pitch on the surface of the graphite block was removed using a grinder. Figure 1 illustrates a schematic of the impregnation apparatus.

### 2.4. Density, Porosity, and Impregnation Rate Measurements

Changes in the densities and porosities of the graphite blocks were measured using the Archimedes method based on ASTM C20 [20]. The impregnation rates of the blocks at various pressures were calculated using Equation (1).
(1)$$\mathrm{Impregnation}\;\mathrm{rate}\;\left(\%\right)=\frac{P-P_{0}}{P_{0}}\times 100,$$
where *P* is the apparent porosity of the impregnated block and *P*_0_ is the apparent porosity of the block before impregnation.

The detailed pore properties of the graphite blocks (e.g., pore size distribution) were evaluated using a mercury intrusion porosimeter (AutoPoreV, Micromeritics, GA, USA).

### 2.5. Contact Angle Measurement of the Impregnation Pitch for Surface Energy Evaluation

The surface tension of the impregnation pitch was calculated by dispersing the pitch in tetrahydrofuran and coating it onto polypropylene film; the solvent was then evaporated in the vacuum oven followed by pressing using an auto-press. The contact angles on the prepared pitch film were measured with 10 μL of distilled water and diiodomethane using a drop shape analyzer (DSA100, Kruss Scientific, NC, USA).

The pitch contact angle on the graphite block was also measured to assess its wetting behavior on the block. Next, 300 mg of the pitch was placed onto the surface of the block, the temperature was raised to 110 °C, and the contact angle was measured after there was no further change in the melted pitch drop shape, as reported in a previous study [21].

## 3. Results and Discussion

### 3.1. Densities and Porosities of the Graphite Blocks before and after Impregnation

The changes in the bulk densities and open porosities of the graphite blocks before and after impregnation are presented in Table 2.

The bulk densities of the graphite blocks increased to 1.80 g/cm^3^ from 1.76 g/cm^3^, and their porosities decreased to 10.34% from 14.12%. However, the densities and porosities of the blocks before impregnation differed slightly—it was difficult to compare the impregnation of each block. Figure 2 depicts the rate of the density increase and impregnation rate of the impregnated blocks with various impregnation pressures from 10 to 50 bar. The density rate increased significantly at impregnation pressures of 40 and 50 bar, whereas the impregnation rate gradually increased as the impregnation pressure increased. When the impregnation pressures were 40 and 50 bar, the impregnation rate significantly increased.

As described previously, a portion of the impregnated pitch can be evaporated upon heating for carbonization, and the impregnated pitch melts during carbonization. These two phenomena usually cause the elution of the impregnated pitch, resulting in a reduction in the effectiveness of the pitch impregnation. The higher impregnation pressures of 40 and 50 bar resulted in less elution of the impregnated pitch. The cause of the reduced pitch elution from the graphite blocks is discussed in later sections.

### 3.2. Calculation of Minimum Impregnatable Pore Size with Various Impregnation Pressures

The pitch elution during the carbonization of the impregnated block depends on the pore size of the graphite blocks. When the impregnated pore size of the block is small enough, the impregnated pitch is difficult to elute from the pores. Therefore, the minimum impregnatable pore size at various impregnation pressures is calculated using Washburn’s equation, as defined by Equation (2) [22,23].
(2)$$P=-\frac{4\gamma \cos\theta }{\delta },$$
where *P* is the pressure, *δ* is the entrance diameter of impregnated pores, *γ* is the surface tension, and *θ* is the contact angle of the impregnator on the graphite block.

In Washburn’s equation, the contact angle of the impregnator on the graphite block (*θ*) can be directly measured by dropping the melted droplet onto the surface of the block, as described earlier. Moreover, the surface tension of the impregnated pitch (*γ*) can be calculated using the Owens–Wendt equation, as defined by Equation (3) [24,25]. As mentioned in the experimental section, the pitch was formed into a thin film, and the water and diiodomethane contact angles on the pitch film were measured five times. The surface tension of the pitch could then be calculated based on the contact angles.
(3)$$\gamma_{L}\left(1+\cos\theta \right)=2{\left(\gamma_{S}^{D}\gamma_{L}^{D}\right)}^{1/2}+2{\left(\gamma_{S}^{P}\gamma_{L}^{P}\right)}^{1/2},$$
where *γ_L_* is the surface tension of the liquid-gas interface, *θ* is the liquid contact angle on the pitch film, $\gamma_{S}^{D}$ is the dispersive part of the solid surface tension, $\gamma_{L}^{D}$ is the dispersive part of the liquid surface tension, $\gamma_{S}^{P}$ is the polar part of the solid surface tension, and $\gamma_{L}^{P}$ is the polar part of the liquid surface tension.

Figure 3 depicts the contact angle measurement results for calculating the minimum impregnatable pore sizes under various impregnation pressures. The average water contact angle on the pitch film was 64.1 ± 7.5°, and the average diiodomethane contact angle on the pitch film was 20.9 ± 4.0°. These contact angles were used to calculate the surface tension of the impregnating pitch of 47–53 dyne/cm. The impregnating pitch contact angle on the graphite block was measured as 133.9 ± 2.7°. With this surface tension and pitch contact angle, the minimum impregnatable pore size ranges of the graphite block at various impregnation pressures were calculated, as depicted in Figure 4. When the impregnation pressures were 10, 20, 30, 40, and 50 bar, the impregnatable pores decreased to 120–150, 60–79, 40–53, 30–39, and 24–31 nm, respectively. The small pores impregnated with the pitch are expected to exhibit less elution of the impregnated pitch, possibly explaining why the higher impregnation pressures, 40 and 50 bar, resulted in less elution.

### 3.3. Pore Properties of Graphite Blocks before and after Impregnation

The pores of the graphite blocks before and after impregnation at various pressures were observed using a scanning electron microscope (SEM); the images are illustrated in Figure 5. The large pores, with diameters more than 400 μm, of pristine block disappear after impregnation. By contrast, no significant differences in the other pores of the graphite blocks occur, even after impregnation at various pressures.

Therefore, a mercury intrusion porosimetry analysis was conducted to investigate the pore properties of the graphite blocks before and after impregnation at various pressures more precisely. The results are depicted in Figure 6.

As depicted in Figure 6a,b, no significant relationship between impregnation pressures and pore size was observed in the large-size pore range of 4500 to 360,000 nm. This may be attributed to the damage of the sample that occurred during sample cutting. The small pore size range of 3 to 100 nm also did not reveal any significant relationship between impregnation pressures and pore size. However, a significant decrease after impregnation in the pore volume of the pore size range of 100 to 4500 nm was observed, as depicted in Figure 6b,c.

Figure 7 depicts the pore size distributions of the graphite blocks before and after impregnation at various pressures. The pore volumes of the pore sizes in the range of 4500 to 360,000 nm increased after impregnation due to the damage to the graphite block samples used in the analyses. Graphitized materials are usually harder than carbonized materials; only carbonization was conducted after the impregnation of the graphite blocks. Therefore, the impregnated blocks featured more weak carbonized parts than the pristine graphite block, resulting in more damage to the impregnated block during the sample preparation.

The pore volumes of the pore sizes in the range of 100 to 4500 nm decreased after impregnation. The relationship between pore volume decrease and impregnation pressure displayed the same trend, as confirmed by the impregnation rates under various pressures. The pore volume with this range significantly decreased after impregnation at 40 and 50 bar. However, the pore volume of pore sizes in the range of 3–100 nm slightly increased after impregnation. Recall from the Washburn’s equation results that when the impregnation pressure is 50 bar, the pores with a size of 24–31 nm can be filled with the impregnating pitch. However, the results depicted in Figure 7 exhibited a slight increase in this pore size range because the thermal contraction of the impregnated pitch occurred during carbonization, resulting in a small amount of mesopore formation.

Figure 8 depicts the mercury intrusion and extrusion curves of the graphite blocks before and after impregnation at various pressures. The hysteresis between mercury intrusion and extrusion is depicted in Figure 8. It is attributed to the presence of ink-bottle-type pores, where the mercury cannot be extruded even after lowering the pressure back to the atmospheric pressure. As depicted in Figure 7, the predominant pore size range impregnated with the pitch was 100–4500 nm. Accordingly, the pitch impregnated into the 100–4500 nm pores was not eluted even after carbonization—it is often difficult to elute impregnated ink-bottle-type pores.

Table 3 presents the pore volumes and ink bottle fractions of the graphite blocks before and after impregnation. Approximately 64% of the pores of the pristine graphite block were ink-bottle-type. Moreover, the decreases in these ink-bottle-type pore volumes were 1.17–5.87%p. Given that the decreases in the porosities of the impregnated blocks were 2.11–3.35%p, the impregnation of the ink-bottle-type pores with the impregnating pitch likely caused them to be filled with pitch without elution during carbonization because they can be narrowed enough to hold the pitch, as depicted in Figure 9.

## 4. Conclusions

Bulk graphite blocks with a bulk density of 1.76 g/cm^3^ were impregnated with pitch under pressures of 10, 20, 30, 40, and 50 bar. The density density increase rates increased by 1.93–2.44%, whereas the impregnation rate calculated from the rate of open porosity decreased by 15.15–24.48%. The density increase rate and impregnation rate were significantly high when the impregnation pressures were 40 and 50 bar. Compared with impregnation pressures of 10, 20, and 30 bar, the minimum impregnatable pore sizes with impregnation pressures of 40 and 50 bar were 30–39 and 24–31 nm, respectively. The mercury intrusion porosimeter analysis results demonstrated that the pressure-sensitive pore size ranges of the graphite blocks were 100–4500 nm. The ink-bottle-type pores in this range contributed the most to the effect of impregnation under pressure because the pitch-impregnated-into-ink-bottle-type pores were difficult to elute during carbonization.

## Figures and Tables

**Figure 1 materials-15-00561-f001:**
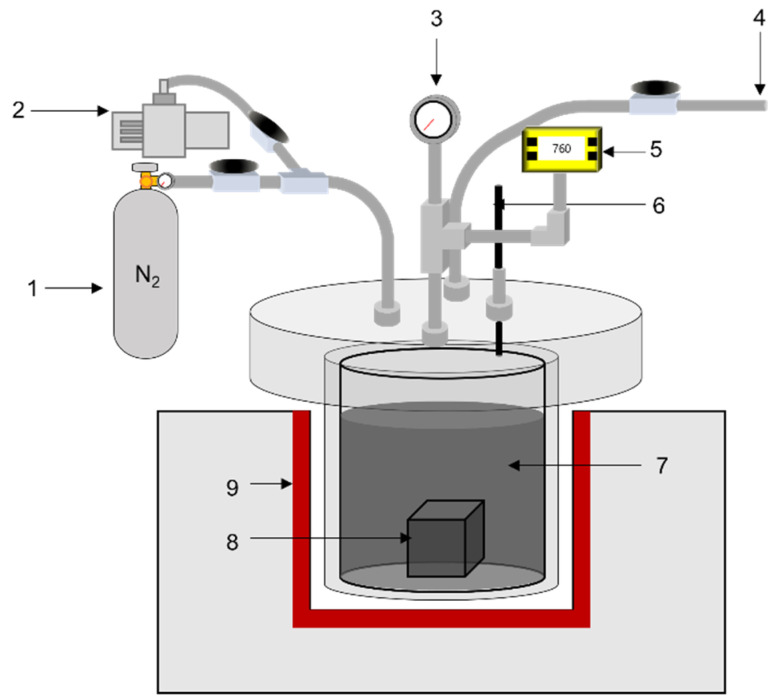
Schematic of impregnation apparatus. (1) N_2_ gas, (2) vacuum motor, (3) pressure gauge, (4) gas outlet, (5) vacuum gauge, (6) thermocouple, (7) pitch, (8) graphite block, (9) heating element.

**Figure 2 materials-15-00561-f002:**
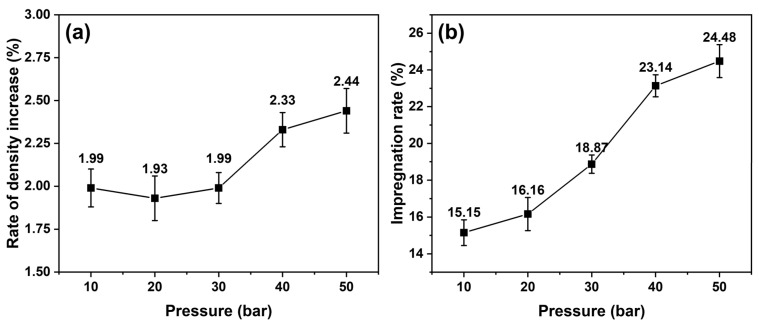
(**a**) Density rate increase and (**b**) impregnation rate of impregnated graphite blocks at various impregnation pressures.

**Figure 3 materials-15-00561-f003:**
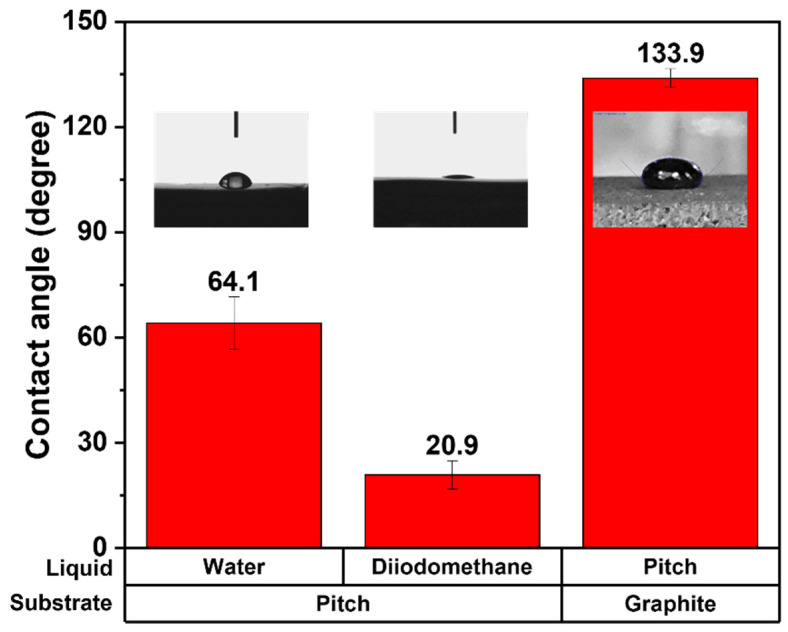
Contact angle measurement results for calculating minimum impregnatable pore sizes under various impregnation pressures.

**Figure 4 materials-15-00561-f004:**
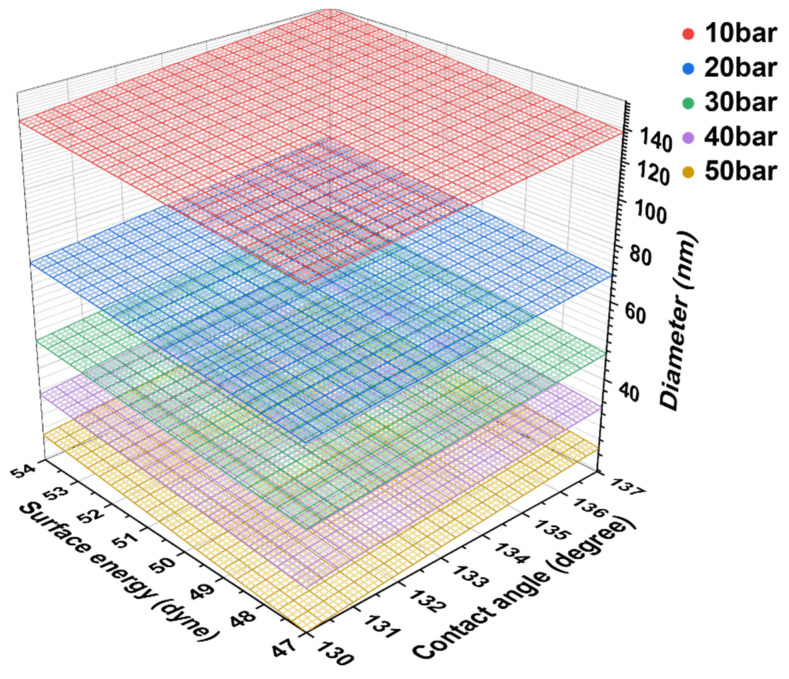
Calculated minimum impregnatable pore size ranges of the graphite block at various impregnation pressures.

**Figure 5 materials-15-00561-f005:**
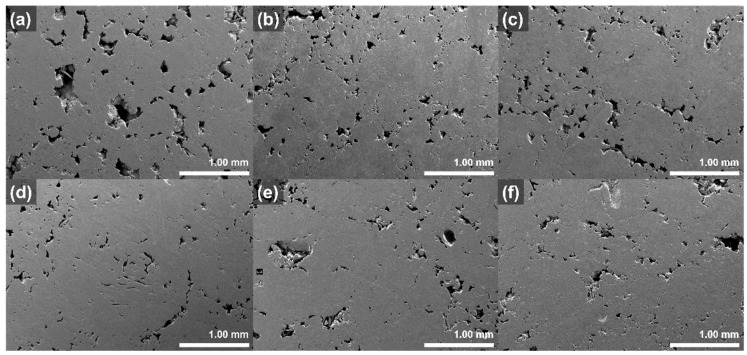
SEM images of graphite blocks before and after impregnation at various pressures: (**a**) pristine, (**b**) 10 bar, (**c**) 20 bar, (**d**) 30 bar, (**e**) 40 bar, and (**f**) 50 bar.

**Figure 6 materials-15-00561-f006:**
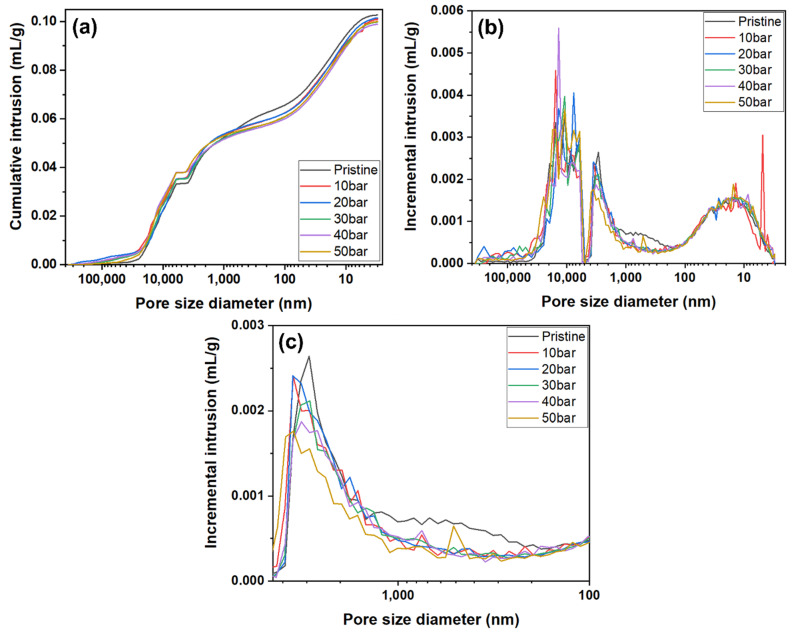
Mercury intrusion porosimetry curves of graphite blocks before and after impregnation at various pressures: (**a**) cumulative intrusion, (**b**) incremental intrusion in range of pore size diameter 360,000 nm to 3 nm, and (**c**) expansion of the incremental intrusion in range of pore size diameter 4500 nm to 100 nm.

**Figure 7 materials-15-00561-f007:**
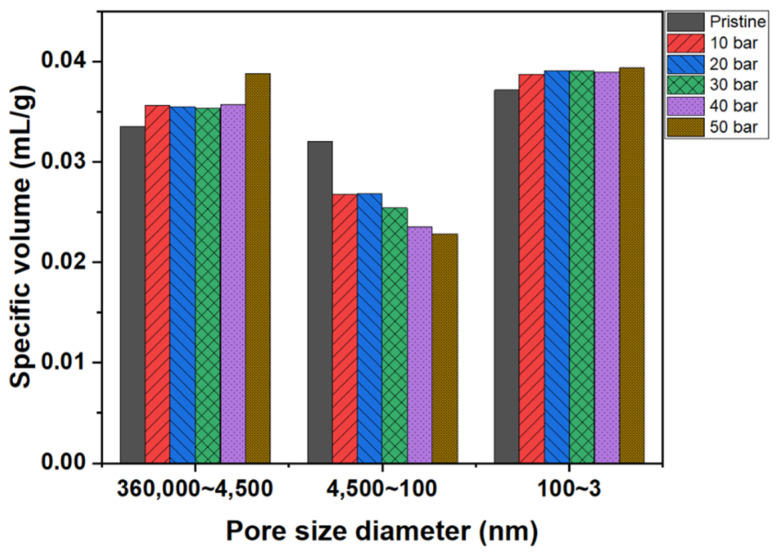
Pore size distributions of graphite blocks before and after impregnation at various pressures.

**Figure 8 materials-15-00561-f008:**
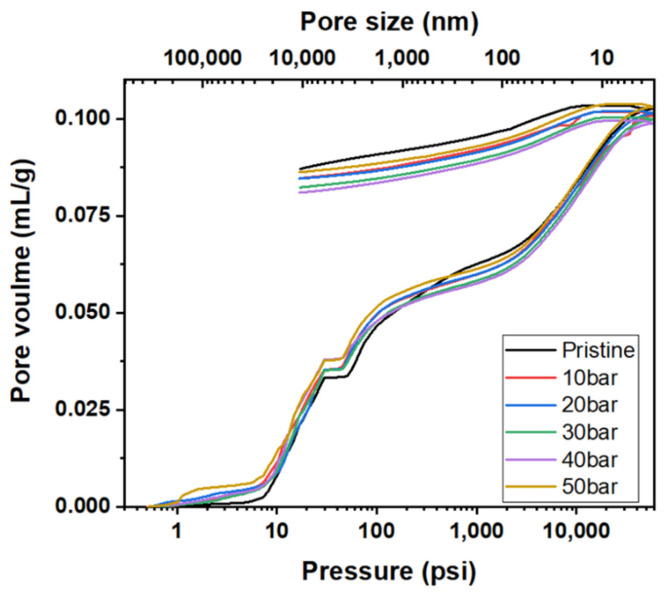
Mercury intrusion and extrusion curves of graphite blocks before and after impregnation at various pressures.

**Figure 9 materials-15-00561-f009:**
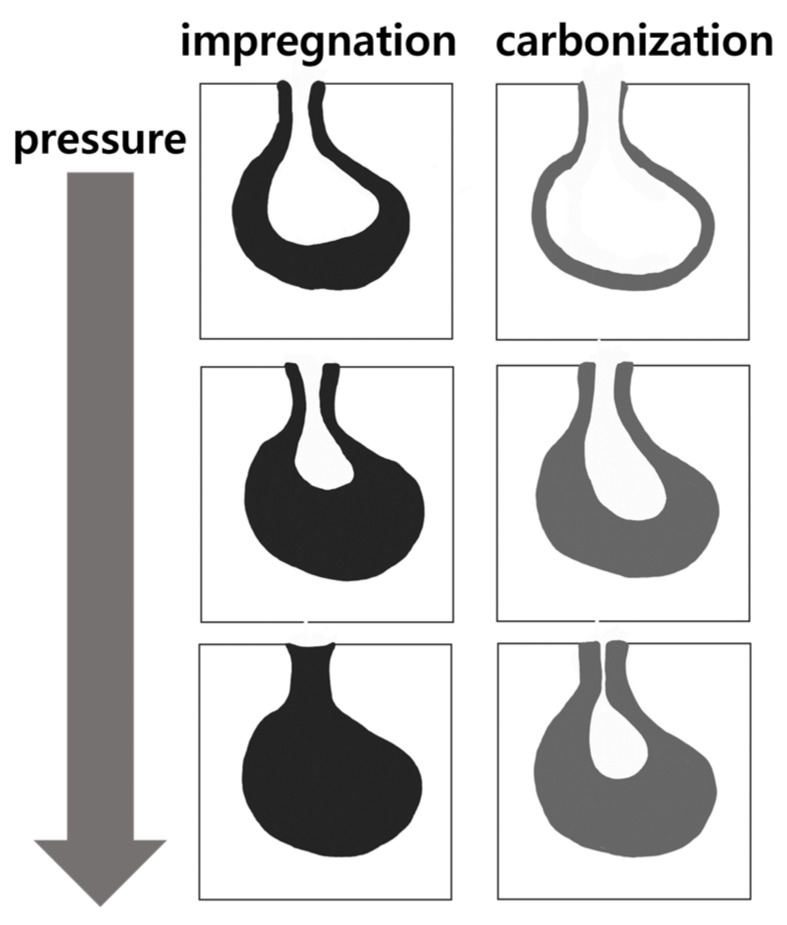
Suggested mechanism of less-impregnated pitch elution from ink-bottle-type pore.

**Table 1 materials-15-00561-t001:** Impregnation pitch specifications.

C (%)	91.87
H (%)	4.53
N (%)	1.24
O (%)	1.36
S (%)	0.64
Carbon yield (%)	39.12
Softening point (°C)	90.00
Quinoline insoluble contents (%)	3.04
Quinoline soluble, Toluene insoluble contents (%)	18.06
Toluene soluble contents (%)	78.87

**Table 2 materials-15-00561-t002:** Bulk densities and open porosities of graphite block before and after impregnation.

Pressure (bar)	Bulk Density (g/cm^3^)		Porosity (%)	
	Unimpregnated	Impregnated	Unimpregnated	Impregnated
10	1.763	1.798	13.95	11.84
20	1.762	1.796	14.12	11.84
30	1.760	1.798	13.93	11.30
40	1.760	1.801	13.93	10.71
50	1.761	1.804	13.69	10.34

**Table 3 materials-15-00561-t003:** Pore volumes and ink bottle fractions of the graphite blocks before and after impregnation.

Impregnation Pressure (bar)	Total Pore Volume (mL/g)	Ink Bottle Pore Volume (mL/g)	Ink Bottle Pore Fraction (%)
pristine	0.10269	0.06573	64.01
10	0.10138	0.06371	62.84
20	0.10114	0.06140	60.95
30	0.09973	0.05907	59.23
40	0.09760	0.05493	59.10
50	0.09997	0.05992	58.14

## Data Availability

The data presented in this study are available on request from the corresponding author.

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
