# Peer review of "Correlation between Pitch Impregnation Pressure and Pore Sizes of Graphite Block"

_materials, 2022, doi:10.3390/ma15020561_

Round 1

Reviewer 1 Report

The authors investigated the effect of the impregnation pressure on the decrease in porosity of the impregnated bulk graphite and revealed the correlation between pitch impregnation behaviors and pores sizes of the bulk graphite. Moreover, this manuscript was well written. Therefore, I would like to suggest this manuscript for publication in this journal after addressing the minor comments.

Q1. It would be useful for the readers if the SEM morphology images of the bulk graphite.

Q2. In addition, how about the mechanical strength of the resultant bulk graphite.

Q3. How to remove excess pitch on the surface of bulk graphite after impregnation? Is there any special step for this? If so, the experimental procedure needs to be included in the revised manuscript.

Q4. The authors should consider an analytical technique to determine the composition or softening points of the impregnation pitch?

Author Response

Comments and Suggestions for Authors

Reviewer 1

The authors investigated the effect of the impregnation pressure on the decrease in porosity of the impregnated bulk graphite and revealed the correlation between pitch impregnation behaviors and pores sizes of the bulk graphite. Moreover, this manuscript was well written.

Therefore, I would like to suggest this manuscript for publication in this journal after addressing the minor comments.

Q1. It would be useful for the readers if the SEM morphology images of the bulk graphite.

=> Based on the reviewer’s suggestion, SEM images were added in Figure 5 (line 193), which illustrates the SEM images of before and after impregnation at various pressure. Line 185–189 describe the related explanations.

“The pores of the graphite blocks before and after impregnation at various pressures were observed using a scanning electron microscope (SEM); the images are illustrated in Figure 5. The large pores, with diameters more than 400 μm, of the pristine block, disappear after impregnation. In contrast, no significant differences in the other pores of the graphite blocks occur, even after impregnation at various pressures”

Q2. In addition, how about the mechanical strength of the resultant bulk graphite.

=> It is challenging to measure the mechanical strength of the resultant bulk graphite because the size of the graphite block used in this study is too small (3 × 3 × 3 cm). We were focused on the impregnation behaviors of the pitch impregnation when the pressures varied. A future study will investigate the mechanical and other properties of the large graphite block with and without pitch impregnation.

Q3. How to remove excess pitch on the surface of bulk graphite after impregnation? Is there any special step for this? If so, the experimental procedure needs to be included in the revised manuscript.

=> After impregnation, excess pitch on the surface of bulk graphite was removed using a grinder. The related sentences have been added to lines 89–90.

“After carbonization of the impregnated block, the eluted pitch on the surface of the graphite block was removed using a grinder.”

Q4. The authors should consider an analytical technique to determine the composition or softening points of the impregnation pitch?

=>Based on the reviewer’s suggestion, the characterization methods of the impregnation pitch are explained in Section 2.2 on lines 68–77.

“2.2 Characterization of the impregnation pitch

The elemental contents of the impregnation pitch were analyzed using an elemental analyzer (Flash 2000, Thermo Fisher Scientific, Waltham, MA, USA). Carbon yield was measured using thermogravimetric analysis (TGA, Q 500, TA instrument, New Castle, DE, USA) from 25 to 1,000 °C with a heating rate of 5 °C/min under a nitrogen atmosphere. The softening point of the impregnation pitch was evaluated based on ASTM D3416 with a heating rate of 2 °C/min using a softening point analyzer (DP-70, Mettler Toledo, Columbus, OH, USA). The contents of the quinoline insoluble were measured based on ASTM D2318, and the contents of toluene insoluble were measured based on ASTM D4312. The properties of the pitch impregnant are presented in Table 1”

Reviewer 2 Report

Dear Authors

In this manuscript, the authors investigate the effect of the impregnation pressure on the decrease in porosity of the impregnated bulk graphite. This manuscript can be accepted after major revision. The following suggestion and comments should be taken:

  1. Generally, the overall English needs to be improved. Please seek guidance from a native English speaker if possible (commas, plural form,  "the" "a", and others could be corrected).
  2. In the introduction. Please add some information about graphite, graphene other graphite/graphene composites and their potential applications. Please add new 2-4 sentences. Please cite (1) J. Mater. Chem. C, 2021, 9, 6722-6748.  DOI https://doi.org/10.1039/D1TC01316E (2) Materials 2021, 14(9), 2448; https://doi.org/10.3390/ma14092448 and (3) Energy Environ. Sci., 2016, 9, 357-390 https://doi.org/10.1039/C5EE02474A
  3. Figure 5. Please correct this image for better quality.
  4. Figure 7. Please correct this image for better quality.
  5. Why authors choose only 1000 oC for the carbonization process? Please explain it (influence of different temperatures).
  6. Could the authors include the standard deviation of the methods?
  7. Could authors add some HRTEM or SEM images of materials?
  8. Could authors add XPS analysis of obtained materials?

Author Response

Comments and Suggestions for Authors

Reviewer 2

Dear Authors

In this manuscript, the authors investigate the effect of the impregnation pressure on the decrease in porosity of the impregnated bulk graphite. This manuscript can be accepted after major revision. The following suggestion and comments should be taken:

1. Generally, the overall English needs to be improved. Please seek guidance from a native English speaker if possible (commas, plural form,  "the" "a", and others could be corrected).

=> The overall English was improved by using the English editing service (Wordvice #134236).

2. In the introduction. Please add some information about graphite, graphene other graphite/graphene composites and their potential applications. Please add new 2-4 sentences. Please cite (1) J. Mater. Chem. C, 2021, 9, 6722-6748.  DOI https://doi.org/10.1039/D1TC01316E (2) Materials 2021, 14(9), 2448; https://doi.org/10.3390/ma14092448 and (3) Energy Environ. Sci., 2016, 9, 357-390 https://doi.org/10.1039/C5EE02474A

=> We added sentences and cited them on lines 31–33. The newly added sentences include the potential applications of graphite, graphene, and graphite/graphene composites.

“Artificial graphite powders such as graphene and graphite/graphene composites are used primarily as anode materials for secondary ion batteries (lithium ion, sodium ion, and aluminum ion) and supercapacitors. In contrast, artificial bulk graphite is used in silicon ingots for semiconductors, nuclear reactor moderators, bearings, special machine parts, mechanical seals, cell divider plates, and steel-making electrode bars [4-9].”  

3. Figure 5. Please correct this image for better quality.

=> Based on the reviewer’s suggestion, we replaced the image with a higher-quality version.

4. Figure 7. Please correct this image for better quality.

=> Based on the reviewer’s suggestion, we replaced the image with a higher-quality version.

5. Why authors choose only 1000 °C for the carbonization process? Please explain it (influence of different temperatures).

=> When the density of graphite block is low, graphitization at a high temperature is required after carbonization. However, because the density of the graphite block used in this study is sufficiently high (about 1.8 g/cm3), only carbonization at 1000 °C was performed without graphitization. Moreover, our future research will analyze the effect of carbonization temperature on the properties of the graphite blocks.

6. Could the authors include the standard deviation of the methods?

=> Based on the reviewer’s suggestion, the standard deviation of the methods added in Figure 2 (line 140).

7. Could authors add some HRTEM or SEM images of materials?

=> Based on the reviewer’s suggestion, SEM images were added in Figure 5 (line 193), which illustrates the SEM images of before and after impregnation at various pressure. Line 185–189 describe the related explanations.

“The pores of the graphite blocks before and after impregnation at various pressures were observed using a scanning electron microscope (SEM); the images are illustrated in Figure 5. The large pores, with diameters more than 400 μm, of the pristine block, disappear after impregnation. In contrast, no significant differences in the other pores of the graphite blocks occur, even after impregnation at various pressures”

8. Could authors add XPS analysis of obtained materials?

=> We appreciate the reviewer’s suggestion concerning XPS analysis. However, XPS is a method for analyzing a small fraction of total material by measuring material surfaces with a depth of less than 10 nm. It is challenging to analyze the impregnated pitch in the graphite block because the graphite block used in this study is bulk graphite (3 × 3 × 3 cm).

Round 2

Reviewer 2 Report

Accept in present form